

# Implementation of a roughness sublayer parameterization in the Weather Research and Forecasting model (WRF version 3.7.1) and its validation for regional climate simulations

Junhong Lee[1,2], Jinkyu Hong[1], Yign Noh[1], Pedro A. Jiménez[3]

[1]Department of Atmospheric Sciences, Yonsei University, Seoul, South Korea
[2]Max Planck Institute for Meteorology, Bundesstraße 53, 20146 Hamburg, Germany
[3]Research Application Laboratory, National Center for Atmospheric Research, Boulder, CO, USA

*Correspondence to*: Jinkyu Hong (jhong@yonsei.ac.kr)

**Abstract.** The roughness sublayer (RSL) is one compartment of the surface layer (SL) where turbulence deviates from Monin–

Obukhov similarity theory. As the computing power increases, model grid sizes approach to the gray zone of turbulence in the energy containing range and the lowest model layer is located within the RLS. In this perspective, the RSL has an important implication in atmospheric modelling research. However, it has not been explicitly simulated in atmospheric mesoscale models. This study incorporates the RSL model proposed by Harman and Finnigan (2007, 2008) into the Jiménez et al. (2012) SL scheme. A high-resolution simulation performed with the Weather Research and Forecasting model (WRF) illustrates the

impacts of the RSL parameterization on the wind, air temperature, and rainfall simulation in the atmospheric boundary layer. As the roughness parameters vary with the atmospheric stability and vegetative phenology in the RSL model, our RSL implementation reproduces the observed surface wind, particularly over tall canopies in the winter season by reducing the root mean square error (RMSE) from 3.1 to 1.8 m s$^{-1}$. Moreover, the improvement is relevant to air temperature (from 2.74 to 2.67 K of RMSE) and precipitation (from 140 to 135 mm month$^{-1}$ of RMSE), although its impact is not as substantial as that to

wind speed. Our findings suggest that the RSL must be properly considered both for better weather and climate simulation and for the application of wind energy and atmospheric dispersion.

## 1 Introduction

The Planetary boundary layer (PBL) is important for the proper simulation of weather, climate, wind energy application, and air pollution. Turbulence plays a critical role in the spatio–temporal variation of the PBL structure through the turbulent

exchanges of momentum, energy, and water between the atmosphere and Earth's surface. Because turbulent eddies in the PBL are smaller than the typical grid size in mesoscale and global models, their impacts must be properly parameterized for atmospheric models. The surface layer (SL) occupies the lowest 10% of the ABL, where the shear-driven turbulence is dominant. In the SL, Monin–Obukhov similarity theory (MOST), which is a zero-order turbulence closure, provides the relationships between the vertical distribution of wind and scalars and the corresponding fluxes in a given stability condition





(Obukhov, 1946; Monin and Obukhov, 1954). The typical numerical weather prediction (NWP) and climate models are applied
for the SL parameterization based on MOST to parameterize the subgrid-scale influences of the turbulent eddies in the PBL
(e.g., Sellers et al., 1986, 1996).

The SL has two parts: inertial sublayer (ISL) and roughness sublayer (RSL). The ISL is the upper part of the SL, where
MOST is valid and vertical variation of the turbulent fluxes is negligible. The RSL is the layer near and within the surface

roughness elements (e.g., trees and buildings). The turbulent transport in the RSL has a mixing layer analogy, and the
atmospheric flow depends on the roughness element properties (Raupach et al., 1996). Accordingly, the flux–gradient
relationships in the RSL deviate from the MOST predictions, and the eddy diffusion coefficients are larger than the values in
the SL (e.g., Shaw et al., 1988; Kaimal and Finnigan, 1994; Brunet and Irvine, 2000; Finnigan, 2000; Hong et al., 2002; Dupont
and Patton, 2012; Shapkalijevski et al., 2016; Zhan et al., 2016; Basu and Lacser, 2017).

Traditionally, the RSL has not been explicitly considered in global and mesoscale models because the PBL in the model
is coarsely resolved, and the lowest model layer is well above the roughness elements accordingly. As the computing power
increases, the regional and global models can be simulated with a finer spatial resolution and the grid size of the NWP model
moving toward the gray zone of turbulence (the scales on the order of the energy-containing range of turbulence). Nevertheless,
studies on the impact of a fine vertical resolution have not been relatively performed. In this perspective, the RSL has an

important implication in atmospheric modelling research. The lowest model layer is typically approximately 30 m high, and
its vertical resolution continues to be better; hence, the models have more than one vertical layer in the RSL, which extends to
2–3 times of the canopy height. Furthermore, model outputs are sensitive to the selection of the lowest model level height
(Shin et al., 2011), but its relation to the RSL has not yet been clearly investigated. Accordingly, turbulent transport in the RSL
must be incorporated particularly in the mesoscale models if the vertical model levels are inside the RSL with an increase in

the vertical model resolution.

The RSL function is a popular and simple method of incorporating the effects of the RSL in the observation and model
(e.g., Raupach, 1992; Physick and Garratt, 1995; Wenzel et al., 1997; Mölder et al., 1999; Harman and Finnigan, 2007, 2008;
de Ridder, 2010; Arnqvist and Bergström, 2015). The RSL function is defined as the observed relationship between the vertical
gradient of wind and scalar and their corresponding fluxes in the RSL. Accordingly, simple relationships are appropriate for

the land surface model in the climate model and for the mesoscale model (Physick and Garratt, 1995; Sellers et al., 1986, 1996).
Despite the importance of the RSL, the Weather and Research Forecasting (WRF) model (Skamarock et al., 2008), which is
one of the widely used models in the operation and research fields, does not consider the effects of the RSL, and has not yet
been evaluated in the regional weather and climate simulations. Harman and Finnigan (2007, 2008) and Harman (2012)
(hereafter, HFs) recently proposed a relatively simpler RSL function that can be used in a wide range of atmospheric models.

The RSL function of the HFs is based on a theoretical background and applicable to a wide range of atmospheric stabilities by
succinctly satisfying the continuity of the vertical profiles of fluxes, wind, and scalars both at the top of the RSL and at the top
of a canopy. The parameterization of HFs has recently been incorporated in a one-dimensional (1D) PBL model and a land
surface model (Harman, 2012; Shapkalijevski et al., 2017; Bonan et al., 2018).





Based on the abovementioned background, this study incorporates the RSL parameterization based on the RSL function of the HFs into the WRF model (version 3.7.1). For this purpose, we reformulate the HFs' RSL parameterization to implement it to the SL parameterization in the WRF model and then discuss the impacts of the RSL parameterization on the regional weather and climate simulations in terms of meteorological conditions near the Earth surface. To the best of the authors' knowledge, our study is the first extensive attempt to incorporate RSL parameterization into the WRF model and to validate it for regional climate simulations. Section 2 is a brief discussion of the RSL parameterization of HFs and the implementation procedures into the WRF model. Section 3 explains the experimental and observational descriptions. Section 4 presents the impacts of the RSL parameterization. Section 5 ends the study with the concluding remarks.

## 2 RSL theory of the HFs

The roughness sublayer parameterization by HFs is adopted herein along with an explanation of the core of the HF model, and the relevant details on this parameterization can be found in Harman and Finnigan (2007, 2008) and Harman (2012). Appendix A lists in alphabetical order the symbols used in this study.

We first define the coordinate alignment for its application to the WRF. The revised MM5 SL scheme in the WRF model defines the vertical origin by the conventional zero-plane displacement height ($d_0$). The same coordinate system is also applied herein. The vertical coordinates $z$ and $\tilde{z}$ in this coordinate system are defined as the distance from $d_0$ and from the terrain surface, respectively; therefore, their relation is $z = \tilde{z} - d_0$. Note that a vertical origin in the HFs is at the canopy height ($h$). MOST says that a variable ($C$), such as wind speed ($u$) and temperature ($T$), has the logarithmic vertical profile:

$$\frac{k}{C_*}(C(z) - C_0) = \ln\left(\frac{z}{z_0}\right) - \psi_c\left(\frac{z}{L}\right) + \psi_c\left(\frac{z_0}{L}\right), \tag{1}$$

where $k$ is von Kármán constant; $C_*$ is a $C$ scale; $C_0$ is $C$ at $z_0$; $z_0$ is the roughness length; $\psi_c$ is the integrated similarity function of $C$; and $L$ is the Obukhov length. The $C$ profile based on the RSL function of the HFs is divided into two layers depending on the relative distance between $h$ and the redefined zero-plane displacement height in the HFs ($d_t = h - d_0$): the upper-canopy layer ($z > d_t$), where the influence of additional mixing by the canopy exists, and the lower-canopy layer ($z < d_t$), where the canopy is the direct source and sink for drag and heat. The vertical profile in the upper-canopy layer is described as follows:

$$\frac{k}{C_*}(C(z) - C_0) = \ln\left(\frac{z}{z_0}\right) - \psi_c\left(\frac{z}{L}\right) + \psi_c\left(\frac{z_0}{L}\right) + \int_z^\infty \frac{\phi_c(1-\hat{\phi}_c)}{z'} dz', \tag{2}$$

where $\phi_c$ is the similarity function of $C$ and $\hat{\phi}_c$ is an RSL function of $C$. The last term in the right-hand side represents the additional mixing caused by the roughness element due to the coherent canopy turbulence, and can be replaced by $\hat{\psi}_c$, which is an integrated RSL function of $C$. The vertical profile from the HFs for the RSL deviates from that of MOST because of $\hat{\psi}_c$, thereby adjusting the logarithmic profile. The $\hat{\phi}_c$ is introduced as follows:

$$\hat{\phi}_c = 1 - c_1 \exp\left[-c_2 \frac{\beta}{l_m} z\right]. \tag{3}$$





The RSL function, $\hat{\phi}_c$ exponentially converges to zero above the RSL. $c_1$ and $c_2$ are then determined from the continuity of $\hat{\phi}_c$ at the canopy top. In the lower canopy layer, $C$ has the following exponential form:

$$C(z) - C_0 = (C_h - C_0) \exp\left(f \frac{z - d_t}{2d_t}\right),$$  (4)

The RSL functions vary with atmospheric stability through $\beta$,

$$\beta = \begin{cases} \dfrac{\beta_N}{\phi_m(z=d_t)} & \dfrac{L_c}{L} > -0.15 \\[2ex] \dfrac{k}{2\phi_m(z=d_t)} + \dfrac{\frac{\beta_N}{\phi_m(z=d_t)} - \frac{k}{2\phi_m(z=d_t)}}{1 + 2\left|\frac{L_c}{L} + 0.15\right|^{1.5}} & \dfrac{L_c}{L} \le -0.15 \end{cases},$$  (5)

where $L_c$ is a canopy penetration depth defined as:

$$L_c = (c_d a)^{-1} = \frac{4h}{LAI}.$$  (6)

where $c_d$ is a drag coefficient at the leaf scale and $a$ is the leaf area density. The parameter $d_t$ and $z_0$ also depend on the stability
because of their dependence on $\beta$:

$$d_t = h - d_0 = \frac{l_m}{2\beta} = \beta^2 L_c,$$  (7)

$$z_0 = d_t \exp\left[-\frac{k}{\beta}\right] \exp\left[-\psi_m\left(\frac{d_t}{L}\right) + \psi_m\left(\frac{z_0}{L}\right)\right] \exp\left[\int_{d_t}^{\infty} \frac{\phi_m(1 - \hat{\phi}_m)}{z'} dz'\right],$$  (8)

where $\psi_m$ is the integrated similarity function for the momentum.

## 3 Incorporation of the roughness sublayer parameterization into the WRF model

The RSL parameterization of the HFs described above is implemented in the Jiménez et al. (2012) revised MM5 surface layer
scheme and Noah land surface model in the WRF (hereafter called the Yonsei University surface layer (YSL) scheme) because
theoretical consistency between the HFs and PBL parameterization. To incorporate the RSL parameterization, it is necessary
to modify the SL scheme and the land surface model as follow (Fig. 1): the first step is to compute the bulk Richardson number
at the lowest model layer, $Bi_b$, by the original equation of Jiménez et al. (2012) [Eq. (9) in their study]:

$$Bi_b = \frac{g}{\theta_a} \frac{\theta_{va} - \theta_{vg}}{[U(z_r)]^2} z.$$  (9)

The second step is to iteratively calculate the atmospheric stability ($z_r/L$) as follows with an accuracy of 0.01:

$$\frac{z_r}{L} = Bi_b \frac{\left[\ln\left(\frac{z_r}{z_0}\right) - \psi_m\left(\frac{z_r}{L}\right) + \psi_m\left(\frac{z_0}{L}\right) + \hat{\psi}_m\right]^2}{\left[\ln\left(\frac{\rho c_p k u_*^{n-1} z_r}{c_s} + \frac{z_r}{z_l}\right) - \psi_h\left(\frac{z_r}{L}\right) + \psi_h\left(\frac{z_l}{L}\right) + \hat{\psi}_h\right]}.$$  (10)

Equation (10) is different from Eq. (23) of Jiménez et al. (2012) by the RSL functions (i.e., $\hat{\psi}_m$ and $\hat{\psi}_h$). After $z_r/L$ is
determined, the third step is to iteratively update $d_t$ and $\beta$ using Eqs. (5) and (6) with an accuracy of 0.0001 because they are





inter-correlated with each other. Subsequently, $z_0$ is iteratively achieved with an accuracy of 0.0001 using Eq. (7) at the given $z_r/L$, $\beta$, and $d_t$. The $u$ profile is determined using Eqs. (2) and (4). Following Jiménez et al. (2012), the profile of a scalar, such as $T$, is determined by

$$\frac{k}{C_*}(C(z) - C_0) = \ln\left(\frac{\rho c_p k u_*^{n-1} z_r}{c_s} + \frac{z_r}{z_l}\right) - \psi_c\left(\frac{z}{L}\right) + \psi_c\left(\frac{z_0}{L}\right) + \int_z^\infty \frac{\phi_c(1-\hat\phi_c)}{z'}dz',\tag{11}$$

for the upper-canopy layer. Equation (4) is used for the lower-canopy layer. Finally, $u_*$ and the aerodynamic conductance ($g_a$)

in the RSL are given to

$$u_* = \frac{ku(z_r)}{\left[\ln\left(\frac{z_r}{z_0}\right) - \psi_m\left(\frac{z_r}{L}\right) + \psi_m\left(\frac{z_0}{L}\right) + \widehat\psi_m\right]} \text{ and,}\tag{12}$$

$$g_{a1} = \frac{\ln\left(\frac{z_r}{d_t}\right) - \psi_h\left(\frac{z_r}{L}\right) + \psi_h\left(\frac{d_t}{L}\right) + \widehat\psi_h\left(\beta\frac{z_r}{l_m}\right) - \widehat\psi_h(\beta\frac{d_t}{l_m})}{ku_*},$$

$$g_{a2} = \frac{S_c}{\beta^2 u_h}\left[\exp\left(\beta\frac{d_t - z_l}{l_m}\right) - 1\right]^{-1},$$

$$g_{a3} = \frac{\ln\left(\frac{\rho c_p k u_*^{n-1} z_l}{c_s} + 1\right)}{ku_*} \text{ and}\tag{13}$$

$$g_a = \frac{1}{1/g_{a1} + 1/g_{a2} + 1/g_{a3}}.$$

## 4 Numerical experimental design

This study evaluated the YSL scheme by making a 1D offline test and a real case simulation. The 1D offline simulation was done to test the YSL scheme performance without feedback to the atmosphere. The two 1D offline simulations are carried out; the YSL, the revised MM5 SL schemes coupled with the Noah land surface model, and the Yonsei University (YSU) PBL

scheme (hereafter offCTL and offRSL experiments). Table 1 presents the idealized data for boundary condition. The real case simulation consisted of two experiments: one-month simulation during winter (January 2016) with the original revised MM5 SL scheme and the YSL scheme (hereafter referred to as the rCTL and rRSL experiments). The rCTL and the rRSL employed the same physics package, except for the SL scheme and the land surface model (Lee and Hong, 2016 and references therein). One-way nesting was applied herein in a single-nested domain with a Lambert conformal map projection to East Asia (Fig. 2).

A 9 km horizontal resolution domain 2 was then embedded in the 27 km resolution domain 1 with 31 vertical layers. The initial and boundary conditions were produced using the National Center for Environmental Prediction Final Analysis data (1° × 1°).

## 5 Observation data for the model validation





The model performance was examined against the surface wind speed and the temperature observed at 46 Automated Synoptic

Observing System (ASOS) sites in Korea (Fig. 2). Quality control of the data includes gap detection, limit test, step test based

on the standard of the World Meteorological Administration and Korea Meteorological Administration (KMA) (Zahumensky,

2007; Hong et al., 2013). For the model validation of the real case simulation, the different measures of the correlation

coefficients, centered root–mean–square differences (RSMD), and standard deviations of the model ($\sigma_m$) normalized by that

of the observation ($\sigma_o$) are shown in a Taylor diagram (Taylor, 2001). In the Taylor diagram, a point nearer the observation at

a reference point (OBS) can be considered to give a better agreement with the observation. We also provide the root–mean–

square error (RMSE) and the mean bias (MB) with the pattern correlation for the rainfall simulation validation.

## 6 Results

### 6.1 Offline simulations

Figure 3 shows the roughness parameters (i.e., $z_0$, $d_t$, and $\beta$) as a function of the normalized atmospheric stability ($L_c/L$) from

the offline simulation of the YSL scheme. The offline simulations reproduced the results of Harman and Finnigan (2007, 2008).

The roughness parameters varied with the atmospheric stability, $L_c/L$, and had peaks at weakly unstable conditions. Note that

the roughness length is constant based on the land cover in the traditional atmospheric model such as the WRF.

Figure 4 indicates that the impacts of the roughness sublayer are also decided by $L_c$, which is a function of $LAI$ and $h$ (Eq.

(6)), thus leading to both diurnal and seasonal variation of canopy roughness. Consequently, the roughness parameters showed

daily and seasonal variations. Overall, the roughness length in the YSL was larger than the revised MM5 SL scheme,

particularly in a smaller $z/L$ (i.e., neutral and unstable conditions) and a larger $L_c$ (i.e., small $LAI$ and/or large $h$). The roughness

length in a stable condition showed relatively smaller changes with $z/L$ and $L_c$ compared to those in the unstable condition.

Our findings suggest that $L_c$ becomes larger in the winter season over tall forest canopies because of the smaller $LAI$, and

higher $h$, thereby leading to relatively larger differences of $z_0$ between the YSL scheme and the default WRF scheme. On the

contrary, a similar value of $z_0$ was observed in summer because of the larger $LAI$. Note that the revised MM5 SL scheme does

not consider $d_t$ and $\beta$.

The RSL function, $\hat{\phi}_c$, introduced by HFs, considers the additional mixing caused by the roughness element. Accordingly,

$\hat{\phi}_c$ should asymptotically converge to the MOST profile (i.e., $\hat{\phi}_c \rightarrow 1$) as $z$ increases with the continuous vertical profiles of the

wind and the temperature. The YSL scheme reproduced these properties of $\hat{\phi}_c$ and matched with the observed profiles inside

canopies: the YSL scheme showed exponential profiles under the canopy top and logarithmic profiles above the canopy top

(Fig. 5). The wind speed and the air temperature above the canopy top were smaller than predicted by MOST because $\hat{\phi}_c < 1$

in the offRSL experiments. Notably, the YSL scheme produced wind and temperature below the zero-plane displacement

height, thereby providing additional useful information on the atmospheric dispersion inside the canopy.



The roughness length changes in the YSL scheme eventually produced changes in the surface energy balance with the
atmospheric stability (Fig. 6). In the offline simulations based on the conditions in Table 1, the YSL produced a larger $z_0$ in
the unstable and near-neutral conditions, but a smaller $z_0$ in $z/L > 3$ compared to the offCTL. The aerodynamic conductance
($g_a$) in the YSL was larger in all the stability conditions even in the stable conditions in which the YSL provided a smaller $z_0$
because the additional term in Eq. (13), $g_{a2}$, dominated over the other effects in the $g_a$ calculation. Accordingly, $H$ and $\lambda E$ in
the YSL were larger than those in the revised MM5 SL scheme. Our finding implies stronger fluxes from the YSL scheme
when the gradient of quantity is the same. However, the impact of the increased $g_a$ was asymmetrical in $H$ and $\lambda E$ depending
on the soil moisture content. In this case simulation, an increase in $\lambda E$ was dominant because the wet condition made more
partitioning of the available energy into the latent heat flux first in the model. However, in the dry condition (i.e., less soil
water content), the YSL produced a larger $H$ without a substantial increase of $\lambda E$ (Fig. S1). A significant increase in $\lambda E$ was
found along with a decrease in $H$ in the strong unstable conditions (Fig. 6) because of the wet soil moisture of 0.25 m$^3$ m$^{-3}$ in
the offRSL simulation in Table 1. The slight increase in the net radiation was mainly associated with the reduced outgoing
longwave radiation caused by the smaller surface temperature in the offRSL.

## 6.2 Real case simulations

Figure 7 shows the real case simulation of the roughness length, 10 m wind speed ($u_{10}$), and 2 m air temperature ($T_2$). We
discuss herein the real cases in the winter season because of stronger effect of the roughness sublayer. The results for the
summer season can be found in the Supplementary Materials. The roughness length in the rCTL experiment was prescribed
from the vegetation data table (i.e., VEGPARM table in the WRF model) and modified by the vegetation fraction (Figs. 2 and
7a).

Overall, the YSL scheme (rRSL experiment) produced 0.2–2.0 m larger $z_0$ than the default values in the rCTL experiment
over the tall canopies, where $L_c$ was large. In contrast, the YSL produced a similar or even slightly smaller $z_0$ over the short
canopies compared to the rCTL experiment. Importantly, the changes of $z_0$ made direct impacts on the momentum fluxes and
thus surface wind speed (Fig. 7b). The typical $u_{10}$ in the rCTL was larger than approximately 3 m s$^{-1}$, and a much stronger
wind ($> 6$ m s$^{-1}$) was observed along the mountains, making a positive bias against the observation. Overall, the YSL scheme
reproduced the better observed diurnal variation by reducing the positive bias of the wind speed (Table 2, Fig. 8). Over the tall
forest canopies, $u_{10}$ in the rRSL was reduced by approximately 30%; however, the region of the smaller wind speed
corresponded to the short canopies, where the roughness length increased (Figs. 7a and b). The YSL scheme particularly
provided better RMSD and correlation coefficient, but less diurnal variability of wind speed because of a relatively larger
reduction of the daytime wind speed (Fig. 8). MB and RMSE decreased from 2.4 m s$^{-1}$ to 1.0 m s$^{-1}$ and from 3.1 m s$^{-1}$ and 1.8
m s$^{-1}$. The Taylor diagram shows that the overall performance of the YSL is better than the default WRF simulation at all the
46 sites. In the Taylor diagram, the statistics moved toward the observation, except for one site, indicating an overall





improvement of 2 m air temperature in the YSL scheme; however, the impact of the RSL was not as large as the wind speed (Table 2, Fig. 9).

Similar to the increases of the aerodynamic conductance in the offline simulations, the YSL in the real case simulation (i.e., the rRSL simulation) simulated a larger $g_a$, particularly in the forest canopies and mountain regions (Fig. 10a). This larger $g_a$ in the YSL led to the increases of the latent heat fluxes by approximately 20 W m$^{-2}$, with an eventual reduction of the soil water content (Fig. 11a). The sensible heat fluxes in the rCTL experiments were generally approximately 80 W m$^{-2}$, except over the snow-covered region where $H$ was approximately 40 W m$^{-2}$. As described in the offline simulation, the changing sign of $H$ in the rRSL depended on the soil moisture content because evapotranspiration is limited in dry soils at given available energy (Figs. 10b and 11b). Consequently, the available energy (=$H + \lambda E$) increased in the YSL scheme, and a larger $\lambda E$ in the rRSL led to a temperature cooler than that in the rCTL experiment (Fig. 7c).

During the winter simulation period, precipitation was observed over an extensive area in the domain, and snow was dominant over the northeastern side of the domain (Figs. 11a and 12). The overall total precipitation in the YSL scheme increased, and the skill score indicated a better simulation of the total amount of precipitation (Table 2, Fig. 12). The pattern correlation of precipitation also increased from 0.972 to 0.978 in the YSL scheme based on 656 rain gauge stations, indicating a better match of the precipitation bands. Despite the increase in $\lambda E$, precipitation decreased in several regions (Figs. 10b and 12b). The differences were not significant in the summer season, and the skill scores in the YSL were similar to the default WRF simulation because our implemented RSL parameterization started to converge to the default WRF in a smaller $L_c$ (i.e., smaller $LAI$ and/or higher $h$) and strong synoptic influences by the summer heavy rainy period (Table S1, Figs. S2–S6).

**7 Summary and conclusion remark**

Turbulent fluxes regulate the planetary boundary layer; thus, they are a crucial process for weather, climate, and air pollution simulations. Most of the NWP and climate models are commonly applied for MOST to compute the turbulent fluxes near the Earth's surface. MOST can be, however, only applicable in the inertial sublayers and the roughness sublayer, the important compartment of the SL, has not been properly parameterized in the model. Increasing the computing power enables us to use more vertical layers in the atmospheric models. Accordingly, the RSL must be incorporated into the model properly to simulate the atmospheric processes in the gray zone. This study proposed the YSL scheme, which incorporated the RSL into the WRF model, based on the RSL model proposed by Harman and Finnigan (2007, 2008) and Harman (2012). We also investigated the impacts of the RSL parameterization on the weather and climate simulations. For these purposes, we designed a series of offline simulations with an idealized boundary condition and a real case simulation to evaluate the performance of the YSL scheme against the observation data.

The offline simulation revealed that the YSL scheme successfully reproduced the features observed in various canopies and Harman and Finnigan (2007, 2008). The RSL function, $\hat{\phi}_c$, asymptotically increased to 1, and the vertical gradients of the wind speed and the temperature decreased in the RSL as $z$ increased, thereby deviating from the MOST prediction. Notably,



unlike the typical assignment of the roughness parameter as a constant, the roughness parameters (i.e., $z_0$, $d_t$, and $\beta$) are functions of the atmospheric stability ($z/L$) and $L_c$. The roughness parameters had a maximum in the weakly unstable condition and in larger $L_c$ (i.e., large $h$ or small $LAI$). In most conditions, the YSL scheme provided a larger roughness length, thereby

producing a wind speed slower than that of the revised MM5 SL scheme. The YSL scheme simulated a colder surface temperature in the unstable conditions.

Meanwhile, the real case simulation showed that the RSL-incorporated WRF produced a larger $z_0$ than the default WRF. This increase in $z_0$ and its change with atmospheric stability eventually made substantial impacts on the surface energy balance, wind, and temperature near the ground surface, momentum transfer, and precipitation. First, an increase of $z_0$ produced larger

momentum fluxes and a smaller 10 m wind speed when the YSL scheme was applied, leading to the mitigation of substantial positive bias in the wind speed in the revised MM5 SL scheme. The larger $z_0$ also made increases in the available energy. This increased available energy is related to the surface cooling caused by the increases in the latent heat fluxes in the wet surface conditions when the RSL parameterization is applied. As a result, these changes in the climate near the ground surface and the surface energy balance regulated precipitation, thereby giving a better simulation of the amount of precipitation and its spatial

pattern.

Our results indicate that the RSL parameterization can be a promising option for resolving the typical overestimation of the surface wind speed of the WRF model, particularly in the tall vegetation and low LAI, despite a relatively larger computing time (e.g., Hu et al., 2010, 2013; Shimada and Ohsawa, 2011; Shimada et al., 2012; Wyszogrodzki et al., 2013; Lee and Hong, 2016). The improvement caused by the RSL parameterization is useful in air quality modelling and wind energy estimation by

better weather and climate in the planetary boundary layer. A further study is necessary to evaluate the characteristics of the YSL scheme in various cases particularly at gray-zone resolutions.

*Code and data availability.* The source code of the Weather Research and Forecasting Model (WRF) is available at http://www2.mmm.ucar.edu/wrf/users/downloads.html. The source code of the YSL scheme and the modelling output presented in this study are available at Github (https://github.com/Yonsei-EAPL/JunhongLee/blob/master/module_sf_ysl.F).

The National Center for Environmental Prediction Final Analysis data that was used as initial and boundary conditions is available at https://rda.ucar.edu/datasets/ds083.2/. The observed wind speed, temperature, and precipitation for the model validation can be downloaded at the Korea Meteorological Administration data portal (https://data.kma.go.kr/cmmn/main.do).





## Appendix A: List of symbols and definitions

| Symbols | Definitions |
| --- | --- |
| $a$ | Leaf are density |
| $Bi_b$ | Bulk Richardson number, at the lowest model layer |
| $c_d$ | Drag coefficient at the leaf level |
| $c_p$ | Specific heat for air |
| $c_s$ | Effective heat transfer coefficient for nonturbulent processes (Carlson and Boland, 1978; Jiménez et al., 2012) |
| $C$ | Variable at $z$, such as $u$ and $T$ |
| $C_0$ | $C$ at $z = z_0$ |
| $C_h$ | $C$ at $h$ |
| $C_*$ | Scale of $C$ |
| $d_0$ | Conventionally defined zero-plane displacement height |
| $d_t$ | Redefined zero-plane displacement height in Harman and Finnigan (2007) |
| $f$ | Parameter related the depth scale of the scalar profile |
| $g$ | Gravitational acceleration |
| $g_a$ | Aerodynamic conductance |
| $h$ | Canopy height |
| $k$ | von Kármán constant |
| $l_m$ | Mixing length for momentum |
| $L$ | Obukhov length |
| $L_c$ | Canopy penetration depth |
| $LAI$ | Leaf area index |



| $p$ | Pressure at $z$ |
|---|---|
| $q$ | Water vapor mixing ratio at $z$ |
| $SW$ | Downward shortwave radiation |
| $S_c$ | Turbulent Schmidt number |
| $S_m$ | Soil mositure |
| $T$ | Air temperature at $z$ |
| $T_2$ | Air temperature at 2 m |
| $T_{sk}$ | Skin temperature |
| $u$ | Wind speed at $z$ |
| $u_{10}$ | Wind speed at 10 m |
| $u_h$ | Wind speed at $h$ |
| $u_*$ | Friction velocity |
| $u_*^{n-1}$ | Previous time step value of $u_*$ |
| $z$ | Height from $d_0$ |
| $\tilde{z}$ | Height from terrain surface |
| $z_0$ | Roughness length |
| $z_l$ | Viscous sublayer depth $=0.001$ (Carlson and Boland, 1978; Jiménez et al., 2012) |
| $z_r$ | Hight of the lowest model layer |
| $z_r/L$ | Atmospheric stability |
| $\beta$ | $u_*/u_h$ |
| $\theta_a$ | Potential temperature of the air at $z_r$ |
| $\theta_{va}$ | Virtual potential temperature of the air at $z_r$ |





| $\theta_{vg}$ | Virtual potential temperature of the air at ground |
| --- | --- |
| $\rho$ | Density of air |
| $\phi_C$ | Similarity function of $C$ |
| $\widehat{\phi}_C$ | RSL function of $C$ |
| $\psi_C$ | Integrated similarity function of $C$ |
| $\psi_h$ | Integrated similarity function of heat |
| $\psi_m$ | Integrated similarity function of momentum |
| $\widehat{\psi}_C$ | Integrated RSL function of $C$ |
| $\widehat{\psi}_h$ | Integrated RSL function of heat |
| $\widehat{\psi}_m$ | Integrated RSL function of momentum |




*Author contribution.* JL and JH contributed to the code development for the YSL scheme, data analysis, and manuscript preparation. YN and PAJ contributed to the writing and editing of the paper and data analysis.

*Competing interests.* The authors declare that they have no conflict of interest.

*Acknowledgements.* This publication was supported by the National Research Foundation of Korea grant funded by the Korean government (MSIT) (NRF-2018R1A5A1024958), the Korea Meteorological Administration Research and Development Program under Grant KMI2018-03512, and the Korea Polar Research Institute (KOPRI, PN19081).



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





**Table 1. Idealized boundary condition for the one-dimensional offline simulation.**

| Variable | Value | Variable | Value |
|---|---|---|---|
| $h$ | 18 m | $S_m$ | 0.25 m$^3$ m$^{-3}$ |
| $LAI$ | 4 m$^2$ m$^{-2}$ | $T(z_r)$ | 300 K |
| Land-use category | Mixed forest | $T_{sk}$ | 303 K |
| $L_c$ | 18 m | $u(z_r)$ | 3 m s$^{-1}$ |
| $p(z_r)$ | 1000 hPa | $u_*$ | 0.5 m s$^{-1}$ |
| $q(z_r)$ | 9.3·10 kg kg$^{-3}$ | $z/L$ | −10–10 |
| $SW$ | 800 W m$^{-2}$ | $z_0$ | 0.25 m |

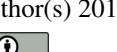



**Table 2. Statistics of the 10 m wind speed, 2 m temperature, and rain rate. The top statistics are presented in bold.**

|  | rCTL | rRSL |
|---|---|---|
| **10 m wind speed** | | |
| Mean bias (m s$^{-1}$) | 2.4 | **1.0** |
| Root–mean–square error (m s$^{-1}$) | 3.1 | **1.8** |
| | | |
| **2 m temperature** | | |
| Mean bias (K) | **−0.92** | −1.16 |
| Root–mean–square error (K) | 2.74 | **2.67** |
| | | |
| **Rain rate** | | |
| Mean bias (mm h$^{-1}$) | −0.018 | **−0.018** |
| Root–mean–square error (mm h$^{-1}$) | 0.194 | **0.187** |
| Pattern correlation | 0.972 | **0.978** |






Input: $\theta(z_r)$, $\theta_{skin}$, $U(z_r)$, $z_r$
Initial: $z_r/L$, $z_0$, $d_t$, and $\beta$

$Bi_b$

$\psi_c(\frac{z_r}{L})$, $\widehat{\psi}_c$

$z_r/L$

Iteration accuracy
of $z_r/L$ : $0.01$
(follow Jimenez et al. (2012))

$d_t, \beta$

Iteration accuracy
of $d_t$ and $\beta$ : $0.0001$

$z_0$

$\psi_c(\frac{z_r}{L})$, $\widehat{\psi}_c$

Iteration accuracy
of $z_0$ : $0.0001$

$u_*$ and $C_h$

**Figure 1: Flow diagram of the RSL parameterization. The gray boxes indicate the iteration module.**



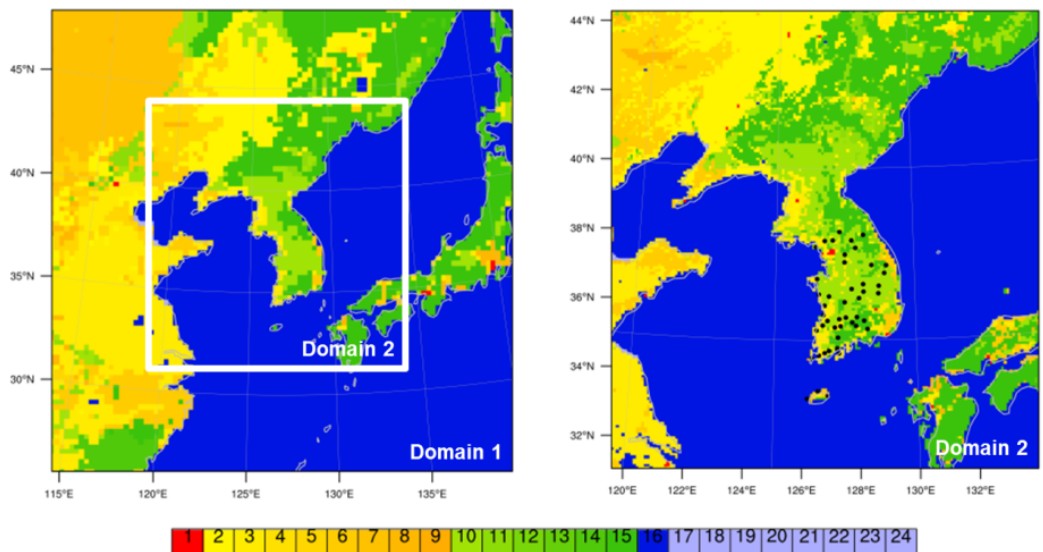

1: urban, 2: dry cropland, 3: irrigated cropland, 4: mixed cropland, 5: cropland+grassland,
6: cropland+grassland, 7: grassland, 8:shrubland, 9: shrubland+grassland,
10: savana:, 11: Deciduous broadleaf, 12: Deciduous needleleaf, 13: Evergreen,
14: Evergreen needleleaf, 15: mixed forest, 16: water,17: Herbaceous wetland, 18: Wooded wetland,
19: Barren/sparsely vegetated, 20~23: tundra, 24: snow/ice

**Figure 2. Domains and land-use category (USGS) of the real case simulation. Black circles denote the automatic synoptic observing system in Korea used for the model validation.**

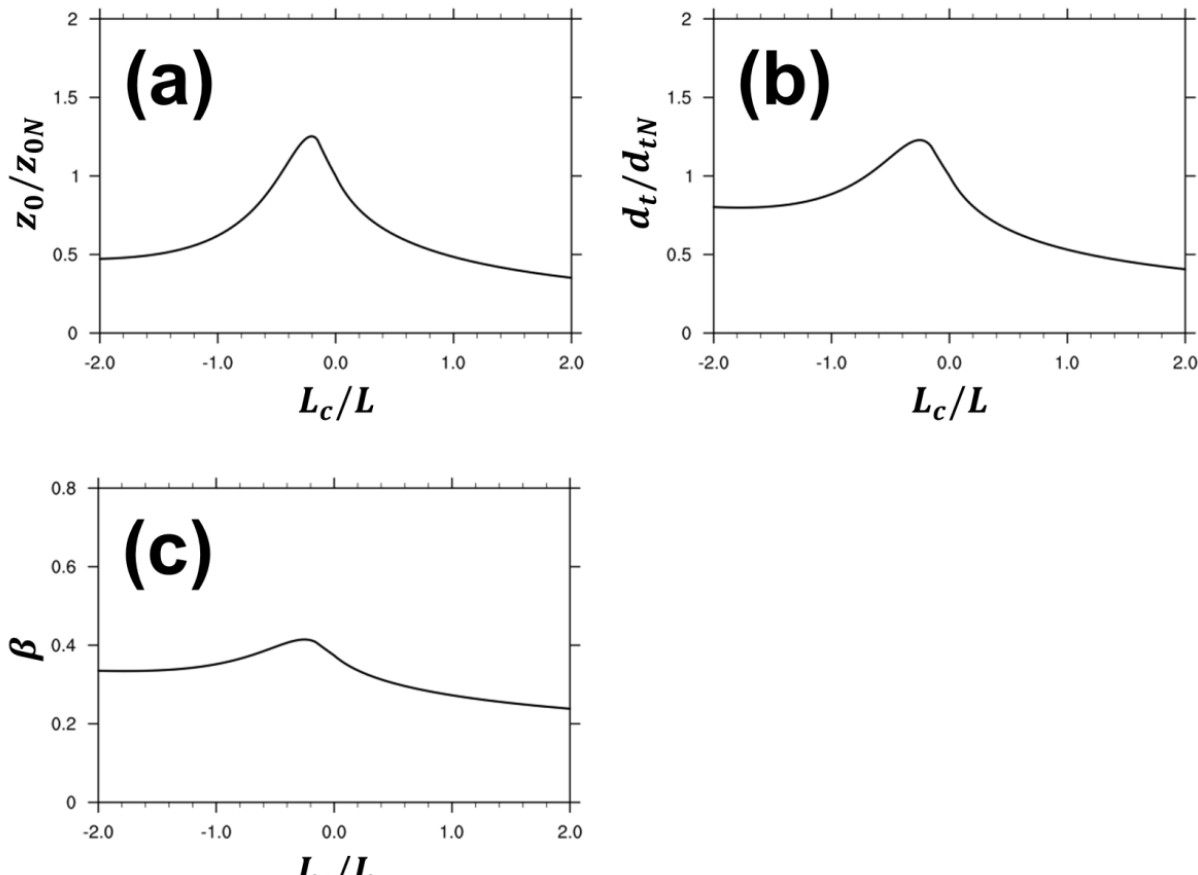

**Figure 3. Roughness length (a), displacement height (b), and $\beta$ (c) normalized by its values in a neutral condition at a given**
**normalized stability ($L_c/L$) from the offline simulation with the YSL scheme.**





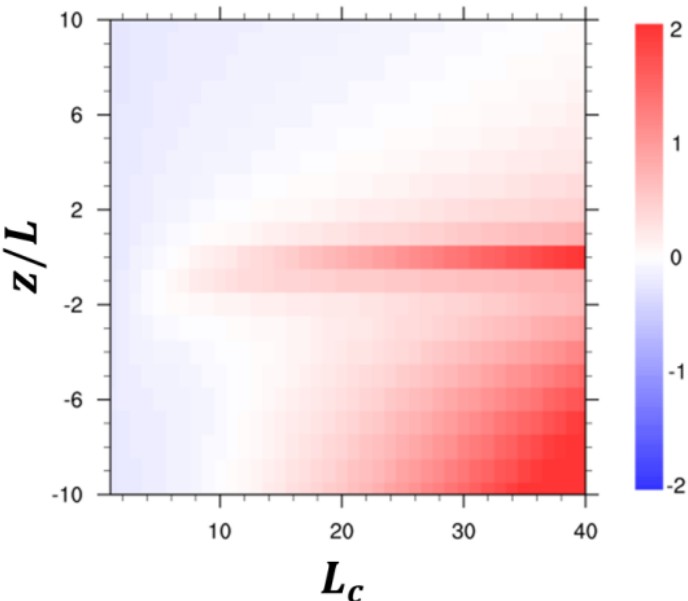

**Figure 4. Roughness length difference (m) between offCTL and offRSL (offRSL − offCTL) at given atmospheric stability ($z/L$) and penetration depth ($L_c$).**




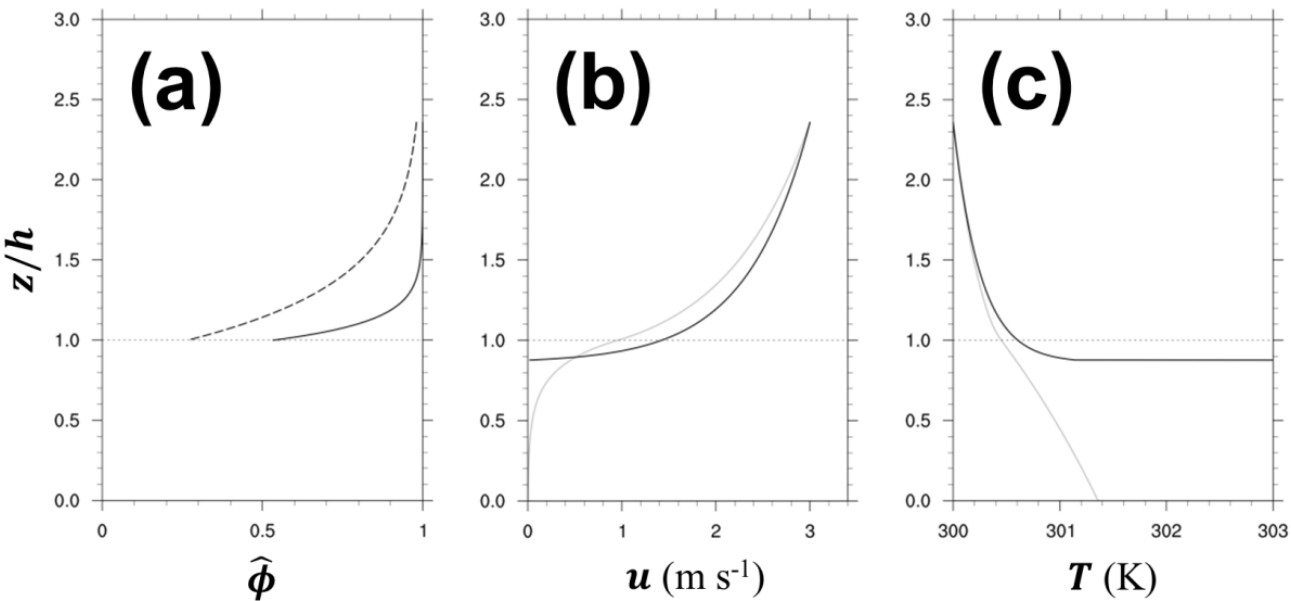

**Figure 5. (a) Profiles of the RSL function for momentum ($\widehat{\phi}_m$, solid line) and heat ($\widehat{\phi}_h$, dashed line), (b) wind speed (m s⁻¹), and (c) temperature (K) at a neutral condition from offCTL (black) and offRSL (gray). The height ($z$) is normalized by the canopy height ($h$).**






**Figure 6. (a) Roughness length (m), (b) aerodynamic conductance (m s⁻¹), (c) sensible heat flux (W m⁻²), (d) latent heat flux (W m⁻²), and (e) net radiation (W m⁻²) at a given atmospheric stability ($z/L$). The black lines denote offCTL, while the gray lines denote offRSL**

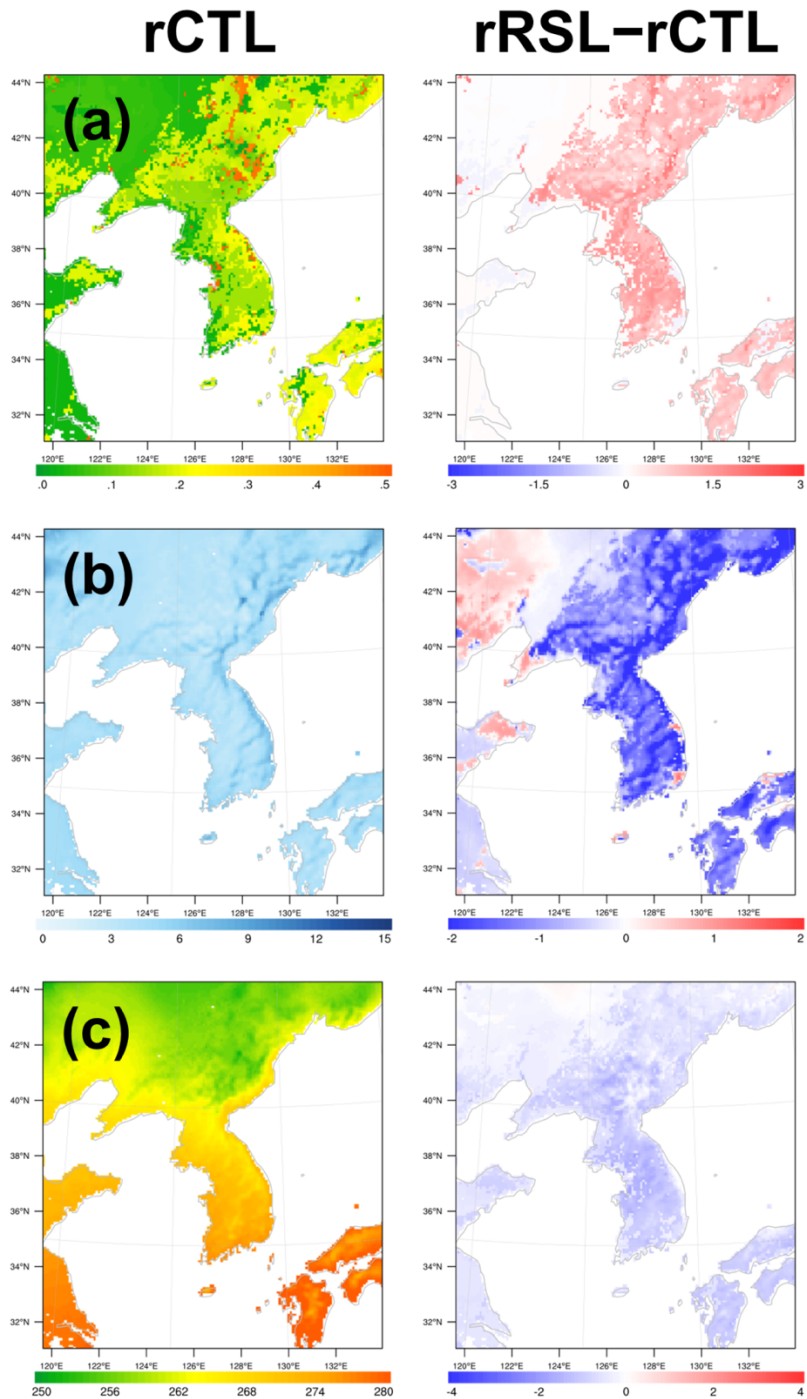

**Figure 7. (a) Roughness length (m), (b) 10 m wind speed (m s⁻¹), and (c) daytime 2 m temperature (K) of the (left) rCTL experiment and (right) the difference (rRSL − rCTL). The results are averaged over a period of one month and masked out over the ocean.**




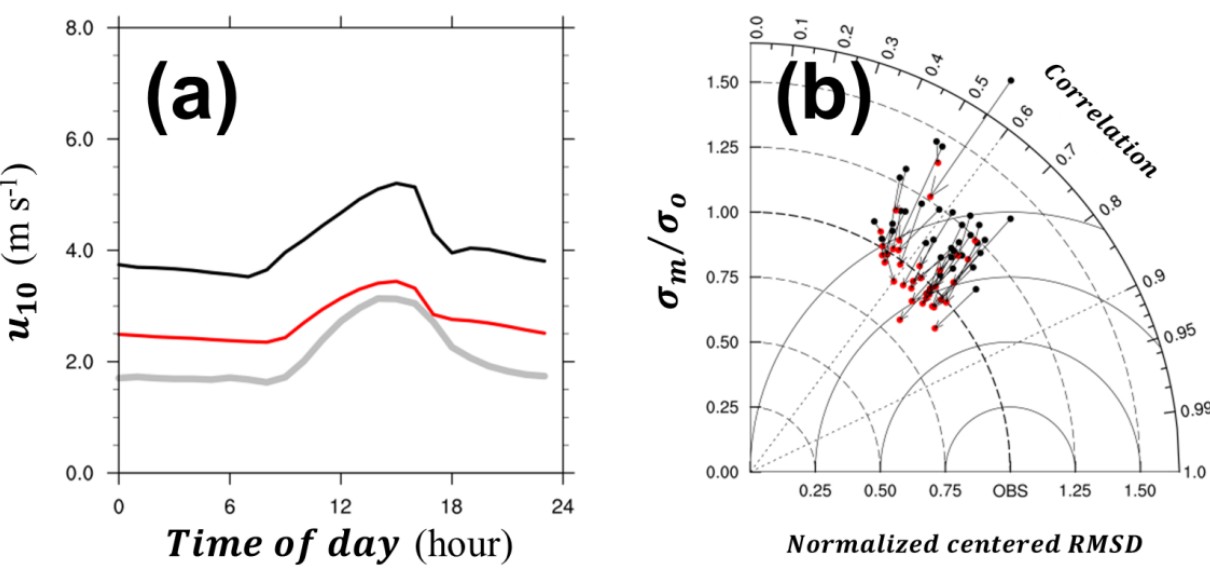

**Figure 8. (a) One month mean diurnal variation of 10 m wind speed and (b) the Taylor diagram showing the correlation coefficient, normalized centered root–mean–square differences (RMSD), and standard deviations of the models ($\sigma_m$) normalized by that of observation ($\sigma_o$) from observation (gray), rCTL experiment (black), and rRSL experiment (red). The vectors indicate the changes of the statistics from rCTL to rRSL. The arrows indicate those from rCTL to rRSL. Every vector shows the movement toward the observation, thereby suggesting the model improvement.**






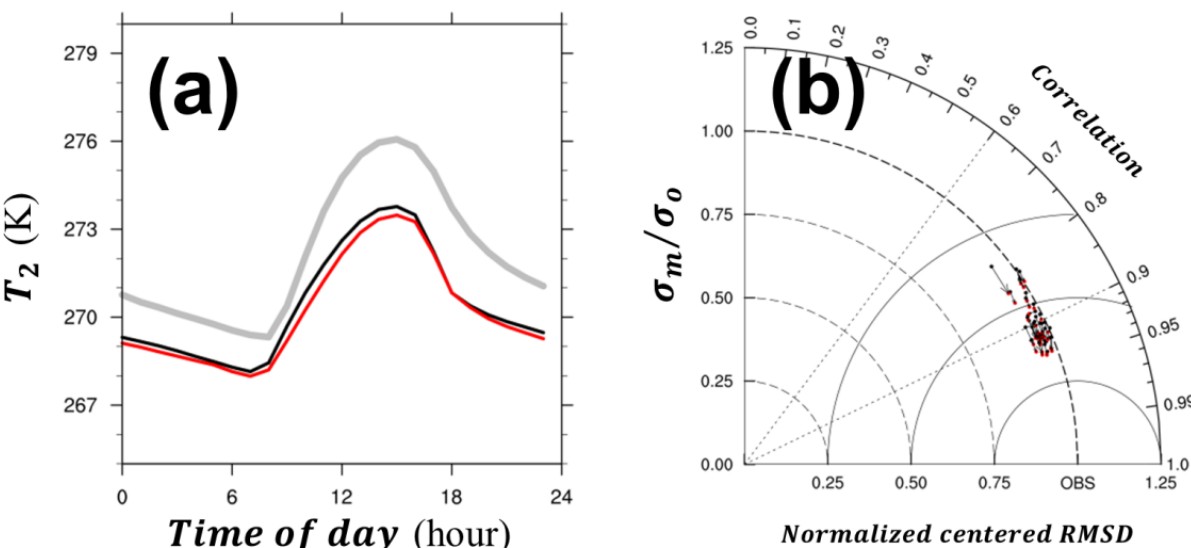

**Figure 9. Same as in Fig. 8, but for the 2 m temperature.**

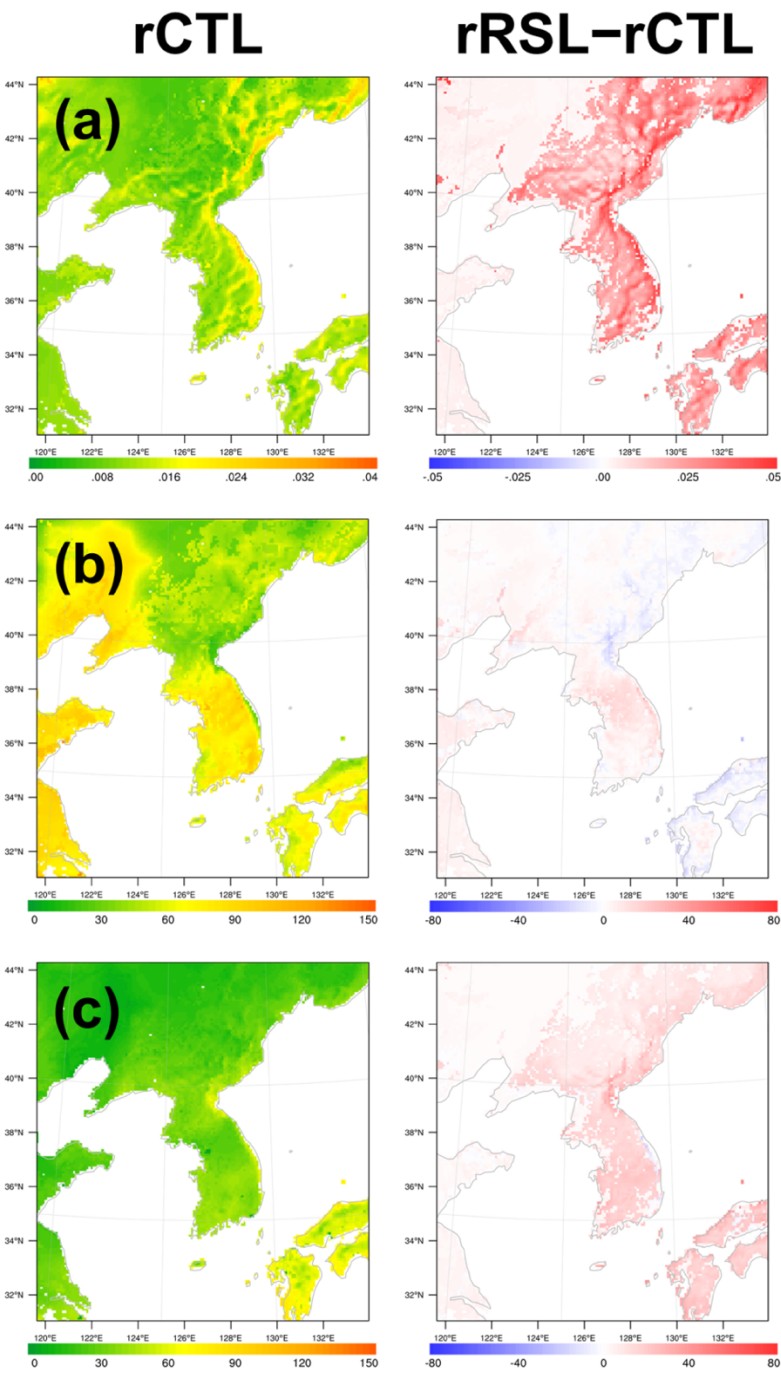

**Figure 10. (a) Aerodynamic conductance (m s$^{-1}$), (b) daytime sensible heat flux (W m$^{-2}$), and (c) daytime latent heat flux (W m$^{-2}$) of the (left) rCTL experiment and (right) the difference (rRSL − rCTL). The results are averaged over a period of one month and masked out over the ocean.**

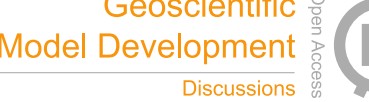



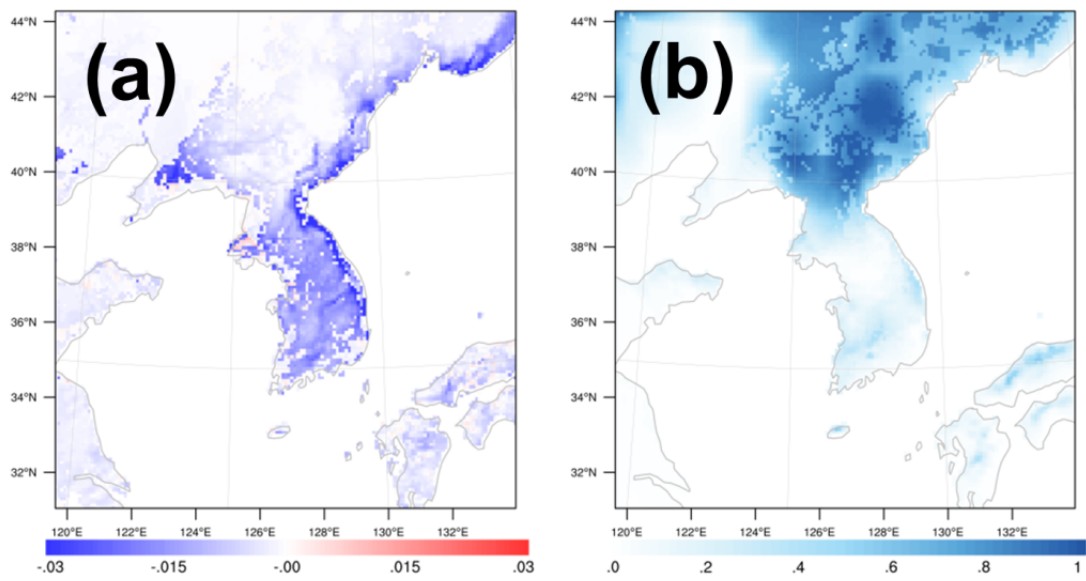

**Figure 11. (a) Difference of the soil moisture (m³ m⁻³) (rRSL − rCTL) and (b) snow cover (%) of rCTL. The results are averaged over a period of one month and masked out over the ocean.**

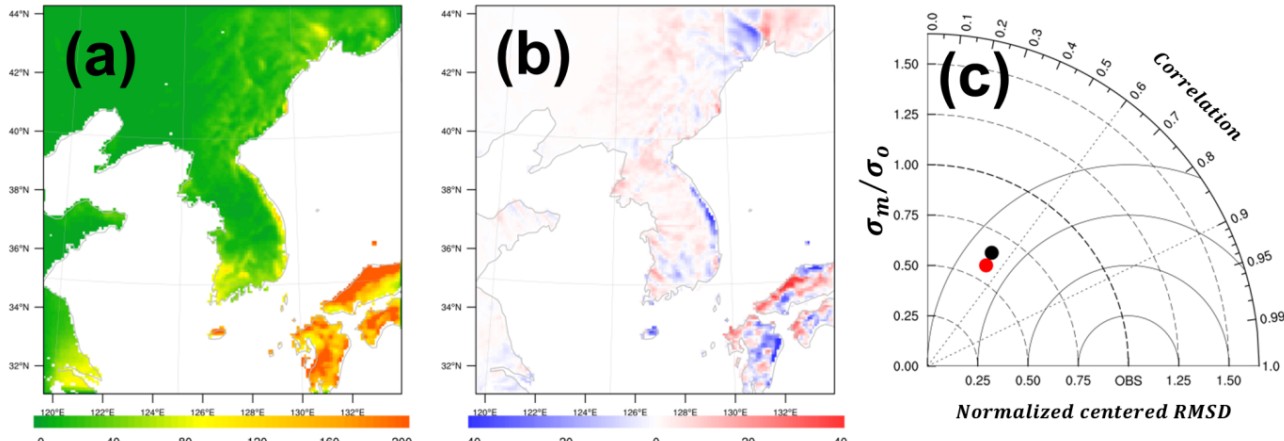

**Figure 12. (a) One month accumulated precipitation of the rCTL experiment (mm) and (b) difference (rRSL − rCTL). (c) Taylor diagram showing the correlation coefficient, normalized centered root–mean–square difference (RMSD), and the standard deviations of models ($\sigma_m$) normalized by that of the observation ($\sigma_o$) and from the rain rate (mm h$^{-1}$) of the rCTL experiment (black) and the rRSL experiment (red) during one month at 656 rain gauges.**