# Peer review of "Table S1. Statistics of the 10 m wind speed, 2 m temperature, and rain rate in July 2016. The top statistics are presented in bold."

_Geoscientific Model Development, 2019_

## Referee Comment (RC1) · Anonymous Referee #1 · 22 Oct 2019

This paper describes the implementation of a roughness sublayer approach into the WRF surface layer physics. The paper describes the method of implementation well and it involves several iterative calculations due to implicit dependencies. Results are shown both for offline and real-data tests. The offline simulations vary the stability, while the real-data simulations show the impact of a wider range of regional variations in surface types for a one-month winter period. The results demonstrate important differences in the wind speed due to the added effective roughness of the new scheme that treats the canopy including forests in a more physically based way following the

methods suggested by Harman and Finnigan (2007,2008).

The paper introduces a useful representation of the roughness sublayer to WRF and is therefore a fitting publication for this journal. It is generally well written, but I will include some technical points that need better description and this could lead to minor revision of the text to improve some explanations.

Technical Points

1. With various z definitions, and d0 and dt, it is easy to be confused about what reference height is being used. The authors should try to ensure consistency, such as when z is referenced and then dt is introduced. Is z still relative to d0 in that case when seen in later equations such as (4)-(5)?

2. Eq. (2). Using an infinite upper bound implies that the length scale in (3) is still below the lowest model level? If so, this needs to be made clear because it is not obvious what length scale (3) has.

3. Eq. (4). This introduces f and does not define it as far as I can tell.

4. The positions of d0, dt and h relative to each other may be helpful to visualize with a schematic, along with how z is defined.

5. line 106. I believe this references Eq. (8) not (7).

6. line 109. ga is introduced without being defined as far as I can tell. This is referred to as aerodynamic conductance but some may be more familiar with it as a surface exchange coefficient for temperature. Is it simply the heat flux divided by the temperature difference? This should be explained.

7. Table 1 shows a z0, but this is probably only in the control experiment as z0 is calculated by the new scheme.

8. With the iterations required, does this add much to the cost of the scheme in computer time.

[Figure]

9. Figure 3. In the idealized case, the control z0 is 0.25 m. Here the figure shows a ratio of z0/z0N. What is z0N so that we can compare it with 0.25 m?

10. Figure 5 is another place where it would have helped to know that the original displacement height is less than the canopy height because we see the CTL values end there.

---

## Referee Comment (RC2) · Anonymous Referee #2 · 28 Oct 2019

Summary.

This paper discusses the addition of a roughness sublayer parameterization in the Weather Research and Forecasting (WRF) model. Following a description of the underlying theory, the implementation is validated in an offline simulation and a real-case scenario over the Korean peninsula. The authors show that the roughness sublayer parameterization leads to model improvements with respect to near-surface wind speed, temperature etc. This paper is of relevance to the research community and potentially for operations, and I encourage its publication following a minor revision.

[Figure]

General comments.

Please check the language used in the paper (grammar and spelling; see also specific comments below).

A key argument for the introduction of the roughness sublayer parameterization in WRF is the increase in computational power and the refinement of the vertical layers in the model. However, in the presentation of the simulations, no reference is made to how many vertical levels are used in the offline or real-case simulations. I encourage the authors to add this information. I would also encourage providing a note on how much more compute time is required in WRF, compared to the original MM5 approach.

Specific comments.

line 12 vs line 26: British or American English? line 12, modelling, is British English; line 26, parameterized, is American English. Please agree on one spelling and make the paper consistent.

line 20: for better weather and climate simulations (s in simulations missing)

line 54: gradient of "wind and scalar" and their corresponding fluxes in the RSL hould that be "wind and scalar variables" ?

line 65-66: "to implement it to the SL parameterization" "to implement it in the SL parameterization" ?

line 68: "to incorporate RSL parameterization" "to incorporate the RSL parameterization"

lines 76-86: this section would benefit greatly from a schematic of the vertical layout (from surface to RSL-ISL interface)

Section 2: all variables that are used need to be introduced, for example: l_m, beta_N, f, phi_m

lines 99-100: "because theoretical consistency" "because of theoretical consistency"

line 105: "iteratively update dt and $\beta$ using Eqs. (5) and (6)" "iteratively update dt and $\beta$ using Eqs. (5) and (7)" ?

line 106: "z0 is iteratively achieved with an accuracy of 0.0001 using Eq. (7)" "z0 is iteratively achieved with an accuracy of 0.0001 using Eq. (8)" ?

line 113-115: please rewrite this sentence and the short sentence in line 115

section 4 - vertical layers used in the experiments?

section 6 - please check the language in particular in this section

line 200 - I would make the discussion of the summer season a new paragraph

line 205-206: check language of this sentence

line 228: "these changes in the climate near the ground surface" - is climate the correct term to use here? also, "ground surface" could be just "ground" or "surface"?

———————————————————

---

## Short Comment (SC1) · 25 Nov 2019

This is an executive editor comment highlighting the ways in which this manuscript is not currently compliant with GMD policy on code and data availability. The following issues need to be resolved in any revised manuscript.

1. Github URLs. Github is an excellent development platform, but it lacks the features required of an archive. GitHub themselves tell authors to use Zenodo for this purpose. The authors should follow the procedure detailed there to

archive the exact version of the software used to create the results presented: https://guides.github.com/activities/citable-code/. The resulting Zenodo repositories present the correct bibliography entries to use.

2. Posting a URL is not a correct way to cite NCEP data. Click on the URL you posted and the very web site you go to provides information on how to cite, including buttons to download the appropriate BibTeX or RIS. Please cite this data properly.

3. The forcing data from the Korea Meteorological Administration is not identified with any precision at all. Please cite the specific data used so that someone who wished to reproduce your work would be able to find the exact data you used. It would also be particularly advantageous if the data could be found without having to read Korean, as I suspect that Korean literacy rates among the wider atmospheric science community are rather low.

Further details on code and data availability requirements are in the GMD model code and data policy: https://www.geoscientific-model-development.net/about/code_and_data_policy.html. The reasons for the policy and more detail are provided in this editorial: https://doi.org/10.5194/gmd-12-2215-2019.

---

## Author Comment (AC1) · 21 Dec 2019

We thank all the reviewers for spending their valuable time to review our manuscdript. We are also happy to receive constructive comments of the reviewers and please check our reponses to your valuable comments below.

Technical Points:
1. With various $z$ definitions, and $d_0$ and $d_t$, it is easy to be confused about what

reference height is being used. The authors should try to ensure consistency, such as when $z$ is referenced and then $d_t$ is introduced. Is $z$ still relative to $d_0$ in that case when seen in later equations such as (4)-(5)?

Reply: As the reviewer pointed out, the coordinate systems between the original surface layer scheme and the RSL scheme proposed by Harman and Finnigan are different. For better understanding of these two different vertical coordinates, we added a figure to show differences of the two vertical coordinates. Please consider that we did coordinate transform from the RSL theory to the WRF surface layer scheme to couple the RSL model into the WRF. That is, $z$ is defined as the distance from the conventional zero-plane displacement height ($d_0$) and therefore, $d_t(= h - d_0)$, distance between $d_0$ and canopy height ($h$) is matched to $z$ at canopy top ($z = d_t$). For better clarification of this point, we revised our manuscript with additional schematic diagram.

2. Eq. (2). Using an infinite upper bound implies that the length scale in (3) is still below the lowest model level? If so, this needs to be made clear because it is not obvious what length scale (3) has.

Reply: About this issue, we want to cite the paragraph in Harman and Finnigan (2007): "the infinite upper bound in (3) indicates that the mixing layer eddies originate at the canopy top hence their influence on the wind speed profile should decrease with increasing height. Therefore the $z \to \infty$ limits of the wind speed profile with and without the roughness sublayer influence ($\hat{\phi}_c \equiv 1$) are equal." Accordingly, the upper bound is not related to the lowest model level and we revised our manuscript for better readability.

3. Eq. (4). This introduces f and does not define it as far as I can tell.

Reply: As the reviewer suggested, we added the definition of f in Appendix A, list of symbols and definitions.

4. The positions of $d_0$, $d_t$ and $h$ relative to each other may be helpful to visualize with a schematic, along with how $z$ is defined.
Reply: As the reviewer suggested, we addd a schematic diagram to describe the coordinate system used in this study.

5. line 106. I believe this references Eq. (8) not (7).
Reply: This is our mistake and we corrected it.

6. line 109. $g_a$ is introduced without being defined as far as I can tell. This is referred to as aerodynamic conductance but some may be more familiar with it as a surface exchange coefficient for temperature. Is it simply the heat flux divided by the temperature difference? This should be explained.
Reply: The reviewer pointed out, we revised the texts with more information on the definitions of aerodynamic conductances.

7. Table 1 shows a $z_0$, but this is probably only in the control experiment as $z_0$ is calculated by the new scheme.
Reply: As the reviewer suggested, we revised the manuscript to clarify this issue.

8. With the iterations required, does this add much to the cost of the scheme in computer time.
Reply: Based on our simulation, the YSL scheme increased the computing time by only 8%. We believe that our scheme is promising because of improvement of meteorology simulation described in our manuscript accordingly. We added this information into the revised manuscript.

9. Figure 3. In the idealized case, the control $z_0$ is 0.25 m. Here the figure shows a ratio of $z_0/z_{0N}$. What is $z_{0N}$ so that we can compare it with 0.25 m?

Reply: z0N is the value of $z_0$ in neutral conditions simulated by the new RSL model and thus Figure 3 shows the dependency of the roughness length with $L_c/L$ by normalizing it with the roughness length in neutral condition. As the reviewer suggested, we revised the texts to clarify the definition of $z_{0N}$.

10. Figure 5 is another place where it would have helped to know that the original displacement height is less than the canopy height because we see the CTL values end there

Reply: As the reviewer pointed out, $z/h$ must be $\tilde{z}/h$ in this figure. We revised the Figures. We appreciate your support for our manuscript and thank you.

---

## Author Comment (AC2) · 21 Dec 2019

We thank all the reviewers for spending their valuable time to review our manuscdript. We are also happy to receive constructive comments of the reviewers and please check our reponses to your valuable comments below.

General comments:
Please check the language used in the paper (grammar and spelling; see also specific comments below). A key argument for the introduction of the roughness sublayer parameterization in WRF is the increase in computational power and the refinement of the vertical layers in the model. However, in the presentation of the simulations, no reference is made to how many vertical levels are used in the offline or real-case simulations. I encourage the authors to add this information. I would also encourage providing a note on how much more compute time is required in WRF, compared to the original MM5 approach.

Reply: As the reviewer suggested, we checked the language in our manuscript. We also revised our manuscript by providing the number of vertical layers and the computing time. In summary, the YSL scheme increased the computing time by only 8% compared to the original MM5 surface layer scheme. We believe that our scheme is promising because of improvement of meteorology simulation described in our manuscript accordingly.

1. line 12 vs line 26: British or American English? line 12, modelling, is British English; line 26, parameterized, is American English. Please agree on one spelling and make the paper consistent.

Reply: We revised the manuscript with consistent English as the reviewer suggested.

2. line 20: for better weather and climate simulations (s in simulations missing)

Reply: We corrected it as the reviewer suggested.

3. line 54: gradient of "wind and scalar" and their corresponding fluxes in the RSL should that be "wind and scalar variables"?

Reply: We corrected it as the reviewer suggested.

4. line 65-66: "to implement it to the SL parameterization" "to implement it in the SL parameterization"?

Reply: We corrected it as the reviewer suggested.

5. line 68: "to incorporate RSL parameterization" "to incorporate the RSL parameterization"

Reply: We corrected it as the reviewer suggested.

6. lines 76-86: this section would benefit greatly from a schematic of the vertical layout (from surface to RSL-ISL interface).

Reply: As the reviewer suggested, we added a schematic diagram and revised our manuscript with a schematic diagram.

7. Section 2: all variables that are used need to be introduced, for example: $l_m, \beta_N, f, \phi_m$.

Reply: Please consider that all variables are defined in Appendix A and we clarified this point in our revised manuscript.

8. lines 99-100: "because theoretical consistency" "because of theoretical consistency"

Reply: We corrected it as the reviewer suggested.

9. line 105: "iteratively update $d_t$ and $\beta$ using Eqs. (5) and (6)" "iteratively update $d_t$ and $\beta$ using Eqs. (5) and (7)"?

Reply: We corrected it as the reviewer suggested.

10. line 106: "$z_0$ is iteratively achieved with an accuracy of 0.0001 using Eq. (7)" "$z_0$ is iteratively achieved with an accuracy of 0.0001 using Eq. (8)"?

Reply: We corrected it as the reviewer suggested.

11. line 113-115: please rewrite this sentence and the short sentence in line 115.

Reply: We revised this sentence as the reviewer suggested.

12. section 4 - vertical layers used in the experiments?

Reply: We provided the information in section 3 and section 4, as the reviewer suggested.

13. section 6 - please check the language in particular in this section.

Reply: We received the English proof service for this section as the reviewer suggested.

14. line 200 - I would make the discussion of the summer season a new paragraph.

Reply: As the reviewer suggested, we made the discussion of the summer season a new paragraph.

15. line 205-206: check language of this sentence.

Reply: We received the English proof service for this section as the reviewer suggested.

16. line 228: "these changes in the climate near the ground surface" - is climate the correct term to use here? also, "ground surface" could be just "ground" or "surface"?

Reply: We revised the sentence as the reviewer suggested.
* * *

---

## Author Comment (AC3) · 21 Dec 2019

We thank all the editor for spending their valuable time to review our manuscdript. We are also happy to receive constructive comments of the reviewers and please check our reponses to your valuable comments below.

General comments:
This is an executive editor comment highlighting the ways in which this manuscript is

not currently compliant with GMD policy on code and data availability. The following issues need to be resolved in any revised manuscript.

Reply: We revised the manuscript to comply with GMD policy on code and data availability. Please check our revision described below and let us know if you have further concern on GMD policy. Thank you very much.

Specific comments:

1. Github URLs. Github is an excellent development platform, but it lacks the features required of an archive. GitHub themselves tell authors to use Zenodo for this purpose. The authors should follow the procedure detailed there to archive the exact version of the software used to create the results presented: https://guides.github.com/activities/citable-code/. The resulting Zenodo repositories present the correct bibliography entries to use.

Reply: We made Zenodo repository of our codes (http://doi.org/10.5281/zenodo.3555537) as the editor suggested and showed this address in the revised manuscript.

2. Posting a URL is not a correct way to cite NCEP data. Click on the URL you posted and the very web site you go to provides information on how to cite, including buttons to download the appropriate BibTeX or RIS. Please cite this data properly

Reply: We revised the manuscript as the editor suggested.

3. The forcing data from the Korea Meteorological Administration is not identified with any precision at all. Please cite the specific data used so that someone who wished to reproduce your work would be able to find the exact data you used. It would also be

particularly advantageous if the data could be found without having to read Korean, as I suspect that Korean literacy rates among the wider atmospheric science community are rather low.

Reply: We provided more details on the observation data in the revised manuscript with a full link address. Please consider the Korea Meteorological Administration provides data on website in Korean only and so in Code and data availability, we showed that the observation data were available upon request to the corresponding author (jhong@yonsei.ac.kr / http://eapl.yonsei.ac.kr).

---

## Author Response (AR2)

**Reply to the editor**

We thank the editor and all the reviewers for spending their valuable time to review our manuscript. As the editor suggested, we replaced "validation" with "evaluation" without any other changes. Thank you very much again for your support.

Sincerely,
Jinkyu Hong

Associate Professor
Department of Atmospheric Sciences, Yonsei University, Korea (Republic of)
Email: jhong@yonsei.ac.kr
Homepage: http://eapl.yonsei.ac.kr